# Rapid Quantitative Detection of Live *Escherichia coli* Based on Chronoamperometry

**DOI:** 10.3390/bios12100845

**Published:** 2022-10-08

**Authors:** Zhuosong Cao, Chenyu Li, Xiaobo Yang, Shang Wang, Xi Zhang, Chen Zhao, Bin Xue, Chao Gao, Hongrui Zhou, Yutong Yang, Zhiqiang Shen, Feilong Sun, Jingfeng Wang, Zhigang Qiu

**Affiliations:** 1School of Environmental and Chemical Engineering, Xi’an Polytechnic University, Xi’an 710600, China; 2Tianjin Institute of Environmental Medicine and Operational Medicine, Tianjin 300050, China; 3School of Marine Science and Technology, Tianjin University, Tianjin 300072, China

**Keywords:** *Escherichia coli*, chronoamperometry, glassy carbon electrode, label-free quantitative detection, electrochemical biosensors

## Abstract

The rapid quantitative detection of *Escherichia coli* (*E. coli*) is of great significance for evaluating water and food safety. At present, the conventional bacteria detection methods cannot meet the requirements of rapid detection in water environments. Herein, we report a method based on chronoamperometry to rapidly and quantitatively detect live *E. coli*. In this study, the current indicator i_0_ and the electricity indicator A were used to record the cumulative effect of bacteria on an unmodified glassy carbon electrode (GCE) surface during chronoamperometric detection. Through the analysis of influencing factors and morphological characterization, it was proved that the changes of the two set electrochemical indicator signals had a good correlation with the concentration of *E. coli*; detection time was less than 5 min, the detection range of *E. coli* was 10^4^–10^8^ CFU/mL, and the error range was <30%. The results of parallel experiments and spiking experiments showed that this method had good repeatability, stability, and sensitivity. Humic acid and dead cells did not affect the detection results. This study not only developed a rapid quantitative detection method for *E. coli* in the laboratory, but also realized a bacterial detection scheme based on the theory of bacterial dissolution and adsorption for the first time, providing a new direction and theoretical basis for the development of electrochemical biosensors in the future.

## 1. Introduction

According to the reports conducted by the World Health Organization, common human illnesses are fundamentally related to the lack of access to safe drinking water and food [1,2]. Pathogens, as the main source of water- and foodborne diseases, have become a major threat worldwide [3,4]. Several pathogenic strains of bacteria are known to cause diverse illnesses and infections, which in some cases can even lead to death [5]. The contamination of water sources with pathogenic bacteria, such as *Salmonella*, *Staphylococcus*, and *Escherichia coli*, can cause typhoid fever, gastroenteritis, diarrhea, cholera, and so on [6]. *E. coli* is one of most important pathogenic bacteria and is widely distributed in nature [7]. It is facultative, anaerobic, Gram-negative, non-sporulating coccobacilli [6]. Considering the pathogenicity and wide distribution of *E. coli* strains in drinking water, food, and river and industrial waters, it is of great significance to develop not only accurate but also rapid methods to detect *E**. coli* in these sources to protect human health [8].

At present, culture and molecular biology–based methods are conventionally used for *E. coli* detection and enumeration [9]. However, these traditional methods are associated with certain disadvantages, such as complicated operation and long detection time (it generally takes 1–2 days to obtain test results); thus, rapid analysis becomes challenging [9]. In recent years, many methods based on different detection principles have been developed, such as PCR [10], immunological detection [11], ATP bioluminescence [12], and flow cytometry [13]. In comparison with traditional culture methods, these new detection technologies have their own advantages, such as short detection time, high sensitivity, and excellent accuracy, but they still have the disadvantage of expensive experimental instruments, high cost, and complex operations. Biosensor-based methods have been established over the past 15 years to accelerate detection speed and enhance detection sensitivity [14,15]. Electrochemical biosensors are one of the most promising tools to detect foodborne pathogens, and accordingly, an increasing number of studies have focused on their application [9]. Some of their advantages, in comparison with other methods, include comparable sensitivity, fast response, ability to analyze a turbid solution, and ability to be miniaturized. In addition, electrochemical biosensors are inexpensive, portable, fast, and easy to operate. However, it is difficult to directly use electrochemical biosensors to detect pathogenic bacteria in untreated samples [9].

Chronoamperometry is an electrochemical detection technique in which step voltage is applied to the working electrode, and a current–time curve is obtained by measuring the current of analyte(s). As compared with other detection methods, chronoamperometry is usually used to quantify known analytes, as it is associated with a better signal-to-noise ratio. The thickness of natural bacterial cell membranes is 5–10 nm [16]. Owing to the selective permeability of the membrane, bacterial resistance is 10^2^–10^5^ Ω cm^2^, and the capacitance is 0.5–1.3 μF/cm^2^. Therefore, the adhesion of bacterial cells to the electrode surface causes changes in the electrochemical reaction speed; chronoamperometry is known for sensitively collecting such signal modulations. Consequently, it is theoretically feasible to use the chronoamperometric method to detect bacterial cells in solution without complex functional modifications. However, as the output signal of the chronoamperometric method is difficult to judge intuitively, it is still necessary to carefully analyze the output electrical signal.

Based on the chronoamperometry method, Foroughi [17,18] calculated the diffusion coefficient of electroactive substances under diffusion control with the Cottrell equation. Jandaghi [19] and Foroughi [20] used the calculation equation of cyclic voltammetry peak current i_p_ to obtain the active surface area of the electrode and took the oxidation peak as the index to determine the actual reaction surface area ratio of the electrode before and after modification. Thus, it is effective and feasible to use the Cottrell equation to analyze the output signal under the condition of diffusion control. At the same time, the detection research based on chronoamperometry still has some defects. Yingdi Zhu [21] realized the specific detection of *E. coli* by the immunoaffinity current method based on redox reaction; however, the detection time was as long as several hours, and the pretreatment of bacterial labeling was needed. Lior Sepunaru [22] used anodic particle coulometry technology and nano-modification to achieve non-specific quantitative detection of *E. coli*; however, this method still demonstrated a detection error that could not be ignored. Based on the principle of bacteria blocking redox reaction, Couto [23] realized the quantitative detection of *E. coli* by analyzing collision frequency; however, this method could not distinguish dead bacteria, and the detection limit was as high as 10^8^ CFU/mL. Therefore, in addition to the detection and analysis methods, the soluble impurities in the solution are also one of the important factors affecting the detection performance.

In this study, based on the detection principle of chronoamperometry, we used an unmodified glassy carbon electrode (GCE) to detect live *E. coli* cells, and the mechanism underlying the effect of live bacterial cells on electrochemical response indices was investigated. Our study complements and perfects the theory of adsorption of bacteria on electrode surfaces under electric fields and realizes the quantitative detection of *E. coli* in solution instead of remaining at the level of a single cell. We believe that our findings will provide a new direction and theoretical basis for the future development of electrochemical biosensors.

## 2. Material and Methods

### 2.1. Chemicals and Reagents

Chemically pure K_4_Fe(CN)_6_·3H_2_O (99.0%) and K_3_Fe(CN)_6_ (99.5%) were purchased from MACKLIN (Shanghai MACKLIN Biochemical Technology, Shanghai, China). The electrode detection solution was composed of 0.2 M KNO_3_ and 1.0 mM K_3_Fe(CN)_6_. An aqueous solution of 20.0 mM K_4_Fe(CN)_6_ was used as the detection solution. All solutions were freshly prepared using ultrapure deionized water (R > 18.2 MΩ) and filtered to remove bacterial contamination.

### 2.2. Bacterial Cells, Growth Conditions, and Intracellular Lysate Preparation

*E. coli* K12-MG1655 (ATCC 47076) was inoculated in Luria–Bertani (LB, Sangon Biotech, Shanghai, China) broth and cultured for 10 h in a shaker (150.0 rpm) maintained at a constant temperature of 37.0 °C (THZ-82A, Tianjin Sateris Experimental Analysis Instrument Company, Tianjin, China). Bacterial suspensions were centrifuged at 4629.0× *g* for 5 min (5804R, Eppendorf, Hamburg, Germany) and washed thrice with phosphate buffered saline (PBS, pH 7.2–7.4) to remove any broth residuals. Finally, bacterial cells were resuspended in PBS, and concentrations were adjusted to 10^8^, 10^7^, 10^6^, 10^5^, 10^4^, 10^3^, and 10^2^ CFU/mL, followed by storage at 4.0 °C until needed; the bacterial concentration was confirmed before electrochemical detection.

Ultrasonic fragmentation was performed to obtain intracellular lysate. Different concentrations of bacterial suspensions were ultrasonically treated with a cell disrupter (SCIENTZ-IID, Ningbo Xinzhi Biotechnology Company, Ningbo, China) at 600.0 W for 15 min in an ice bath, followed by filtration through a 0.22 μm filter membrane to remove membrane fragments. The filtrates were subsequently stored at 4.0 °C.

Inactivated bacterial cells with an intact cell structure were obtained via glutaraldehyde fixation. Different concentrations of bacterial cell precipitates were prepared by centrifugation for 5 min at 4629.0× *g*, followed by fixation in 2.5% glutaraldehyde (Phygene, Fujian, China) for 24 h at 4.0 °C. The inactivated cells were then centrifuged at 1157.0× *g* for 15 min and washed thrice with PBS to remove any residual fixative solution. Finally, the cells were resuspended in PBS and stored at 4.0 °C until needed.

### 2.3. Electrochemical System

Electrochemical measurements were performed using an electrochemical workstation (CHI660E, Beijing Huake Putian Technology Company, Beijing, China) in a three-electrode system. A GCE (diameter of 3.0 mm) served as the working electrode, a platinum wire electrode as the auxiliary electrode, and an Ag/AgCl electrode as the reference electrode; the electrolytic cell volume was 10.0 mL. All electrodes were purchased from CHI, Beijing Huake Putian Technology Company, China.

Before electrochemical measurements, the working electrode (GCE) was polished with alumina powder with different particle sizes (1.0, 0.3, and 0.05 μm) successively and then cleaned using 1.0 M HNO_3_, 1.0 M NaOH, acetone, ethanol, and ultrapure water in an ultrasonic bath (KQ-100E, Kunshan Ultrasonic Instrument Company, Jiangsu, China) at 100.0 W for 5 min. The electrode was then activated by cyclic voltammetry in 0.5 M H_2_SO_4_ [24]. As for the activation parameters, the scanning range was −1.0 to 1 V.0 and the scanning speed was 500.0 mV/s; cyclic voltammetry measurements were obtained for 100 cycles. Further, to test the performance of the GCE, cyclic voltammetry measurements were obtained for another 20 cycles at −0.2 to 0.6 V and 50.0 mV/s. When the ΔEp was <80.0 mV and peak current ratio was approximately 1:1, the GCE was used for detection purposes.

### 2.4. Electrochemical Measurements

Bacterial suspensions (1.0 mL) were added to the detection solution (5.0 mL), gently mixed, and then transferred to the electrolytic cell. When the open circuit potential was stable, chronoamperometry measurements were obtained, using three independent tests for each sample. The electrochemical parameters were as follows: the initial potential was 0.6 V (vs. Ag/AgCl), measurement time was 200 s, sampling interval was 0.1 s, and standing time was 2.0 s. The current signal was recorded for analyses. After every test, the solution remaining on the GCE surface was gently rinsed with ultrapure water. All experimental procedures were performed in a sterile environment.

### 2.5. Data Analysis

Output data from the electrochemical workstation control software was obtained in Excel format and analyzed using the data module of Origin (Origin 2020, OriginLab, Northampton, MA, USA).

The original data of the *i-t* curve was fitted with Formula (1) [25]:(1)$$i=i_{0}+\frac{E}{R_{s}}\cdot e^{-\frac{t}{R_{s}\cdot C_{d}}}$$
where *E* = step voltage, *t* = time, *R_s_* = equivalent resistance of the external circuit, *C_d_* = equivalent capacitance of the electric double layer, and *i*_0_ = stable current. *i*_0_ can be calculated using the fitting curve. When the charging process reached equilibrium, *i*_0_ could be expressed as follows:(2)$$i_{0}=\frac{E}{R_{s}}$$
where *E* = step voltage, and *R_s_* = equivalent resistance of the external circuit. Bacterial cell adhesion to the electrode surface may affect *R_s_*, so *i*_0_ was selected as an indicator of electrochemical response.

Integration has a smoothing effect on random noise in transient currents. The electric quantity *Q* is the integral of the current I [25]. The *i-t* curve was integrated to obtain the *i-q* curve, which was fitted using the following equation:(3)$$Q=\frac{2nFSD_{0}^{\frac{1}{2}}C_{0}^{*}t^{\frac{1}{2}}}{\pi^{\frac{1}{2}}}+Q_{dl}+nFS\Gamma_{0}$$
where *n* = number of electrons involved in the reaction, *F* = Faraday constant, *S* = actual electrode area, *D_O_* = diffusion coefficient of substance O, *C_O_^*^* = bulk concentration of substance O, *t* = time, *Q_dl_* = electricity required to charge the electrode double layer, and *nFSГ_O_* = Faraday component of surface adsorption O reduction.

Formula (3) is the chronographic electric quantity curve when the electrolyte remains stationary in homogeneous phase, where 2*nFSD_0_*^1/2^*C_0_^*^t*^1/2^*/Π*^1/2^ represents the amount of electricity required for the electrochemical indicator to diffuse to the electrode surface for oxidation or reduction within a certain detection time. *Q_dl_* represents the electric quantity required for GCE double-layer charging, and *nFSГ*_0_ represents the electricity required for the oxidation of K_4_Fe(CN)_6_ adsorbed on the surface.

*A* was calculated as follows:(4)$$A=\frac{2nFSD_{0}^{\frac{1}{2}}C_{0}^{*}}{\pi^{\frac{1}{2}}}$$

The actual electrode area *S* was a variable, the rest were constants [25]. Bacterial cell adhesion to the electrode surface may affect *S*, so *A* was selected as another indicator of electrochemical response.

The two electrochemical response indices *i*_0_ and *A* were assessed under different bacterial concentrations, and equations to determine the relationship between them were established.

### 2.6. Analysis of Spiked Samples

Different concentrations of *E. coli* suspensions in phosphate buffer solution (PBS) were obtained, followed by plate counting and electrochemical measurements. The plate counting data were compared with the electrochemically measured *E. coli* concentration results to analyze differences between them. The plate counting results served as the standard, and the error range of the electrochemical detection method was calculated.

### 2.7. Changes in E. coli Concentration in the Detection Solution

The plate counting method [26,27] was used to detect *E. coli* concentration in the detection solution before and after electrochemical testing. The changes in bacterial concentrations were calculated using the following formulae:(5)$$\Delta E=\left(C_{2}-\frac{C_{0}+C_{1}}{2}\right)\times V$$
(6)$$\Delta C=\frac{C_{2}-\left(C_{0}+C_{1}\right)∕2}{\left(C_{0}+C_{1}\right)∕2}\times 100\%$$
where *C*_0_ = *E. coli* concentration in the control electrolyte, *C*_1_ = *E. coli* concentration in the electrolyte before testing, *C*_2_ = *E. coli* concentration in the electrolyte after testing, *V* = electrolyte volume, *ΔE* = changes in *E. coli* quantities in the electrolyte before and after testing, and *ΔC* = percentage change in *E. coli* concentration in the electrolyte.

### 2.8. Fluorescence Microscopy

After electrochemical testing, the working electrode (GCE) was immediately removed from the detection solution, and its surface was gently washed using sterilized PBS to remove any residual detection solution and non-adsorbed bacterial cells. A DAPI kit (DAPI, Sangon Biotech, Shanghai, China) was used for staining *E. coli* cells on the electrode surface, according to manufacturer’s instructions, followed by observation under a fluorescence microscope (Olympus BX51, Olympus Corporation, Tokyo, japan). The excitation wavelength was 340.0 nm, and the emission wavelength was 488.0 nm. The GCE that detected electrolytes without bacteria served as the blank control.

### 2.9. Scanning Electron Microscopy

The pretreatment method was similar to the aforementioned method. After electrochemical testing, the working electrode (GCE) was immediately removed from the detection solution, and its surface was gently washed with sterilized PBS. Glutaraldehyde was used to fix bacteria cells adsorbed on the electrode surface; the fixation solution was added dropwise on the electrode surface, followed by incubation at 4.0 °C for 24 h. Subsequently, any residual solution was rinsed away with sterilized PBS. The electrode was then dried and observed under a scanning electron microscope (KYKY-EM6200, Zhongke Science Instrument, Beijing, China).

### 2.10. Fulvic Acid Sample

Fulvic acid (FA) was selected to simulate dissolved organic matter in the river. The detection sample solution was replaced with PBS solution containing 10 mg/L FA; the other experimental conditions remained unchanged.

### 2.11. Statistical Analysis

All experiments were conducted independently in biological triplicate. Curve fitting used the Levenberg-Marquardt algorithm.

## 3. Results and Discussion

### 3.1. Establishment of the Detection Method

Herein we report a quantitative method involving the use of a GCE to detect live *E. coli* cells (Figure 1). As the selected *E. coli* strain carries a negative charge on its cell membrane surface when pH is greater than 4.4, and the electrolyte pH was 7.15, owing to electrotaxis [28], it should move toward the anode by electrophoresis in a direct current electric field [29]. Because the galvanotaxis threshold of *E. coli* depends on solution conductivity [28], the migration rate of *E. coli* in the electric field is bound to remain stable under constant electrolyte concentration and constant voltage. In this study, a GCE was used as the anode, and electrolytes remained the same. *E. coli* was observed to migrate from the electrolyte to diffusion layer on the GCE surface (Figure 1A), and it was finally adsorbed on the electrode surface as a result of electrostatic interaction between the charge carried by *E. coli* cells and the micropores on the GCE surface [30] as well as covalent bonding between the functional groups on the surface of *E. coli* cells and the rich functional groups on the GCE surface [31]. The pH of the test solution was 7.15, and there is no evidence that a neutral pH would have a significant effect on the functional groups and the charge on the electrode surface. We believe that under this detection condition, the electrostatic force between the negative charge of *E. coli* and the positive charge of GCE with the anodic polarization potential would be the key factor to cause bacterial adsorption. *E. coli* adsorption on the GCE surface sealed the electrode and inhibited the reaction, i.e., the oxidation of Fe(CN)_6_^4−^ (Figure 1B). In this manner, changes in the effective area of the electrode can be recorded by detecting changes in electrochemical parameters, obtaining an estimate of the number of adsorbed *E. coli* cells.

First, we established a method for the electrochemical activation of the GCE (Appendix A). As mentioned above, we obtained cyclic voltammetry measurements over 100 cycles (refer to Section 2.3). The electrochemically activated GCE not only eliminated bacteria and impurities on its surface but also promoted the formation of a weak porous film, owing to the rupture of some crosslinks between polyaromatic rings on the glassy carbon surface [32]. The electrode surface became rough, and the microscopic surface became larger, but the effective surface did not show a significant change.

Second, K_4_Fe(CN)_6_ was included in the detection solution as a source of electroactive ions, which had the advantages of good reversibility and high stability; it also showed fast electron transfer ability to generate sensitive electrochemical signals on the electrode [33]. To maintain the osmotic pressure and activity of *E. coli*, 20.0 mM K_4_Fe(CN)_6_ was used as the electrolyte [34].

Third, the step voltage was set to 0.6 V (vs. Ag/AgCl). The reaction rate of Fe(CN)_6_^4−^ ions on the electrode surface at 0.6 V (vs. Ag/AgCl) was adequately fast; the entire electrochemical process was controlled by diffusion (Appendix A). In a previous study, a single *E. coli* cell could be detected on the surface of an ultramicroelectrode by using the step voltage of 0.6 V while avoiding damage to bacterial cells [34]. However, that study only explained the phenomenon and could not quantitatively detect a large number of *E. coli* in the test sample; our method made up for this deficiency. Voltage as low as 0.6 V is unlikely to have a substantial impact on bacterial activity, particularly as the response current was well below the minimum value lethal to bacteria (25 mA) [35].

Finally, detection time was another important parameter. If the detection time is too short, the current may not reach a stable state; however, if it is too long, *E. coli* cells adsorbed on the electrode surface may form clusters. Moreover, it is difficult to prolong the measurement time beyond 300 s [25]. Thus, the measurement time used herein was set at 200 s (Appendix A).

During the whole testing process, accumulation of cells was the key factor affecting the accuracy of this technique; this took place from the beginning to the end of the detection. With the application of step voltage, bacteria continuously migrated and adhered to the electrode surface within the direct current electric field. With the increase of detection time, the number of bacteria continually accumulated on the electrode surface, which resulted in a change in the level of the current. The quantitative detection of bacteria was based on analyzing of the change in current. After we carried out a series of experiments, the optimal detection time (Appendix A) was determined, and the detection time was unified under all concentrations to prevent the difference of cumulative time affecting the results.

### 3.2. Quantitative Detection of E. coli by Monitoring i_0_ and A

As no shielding device was used in the electrochemical measurement, the detected current signal was considerably affected by environmental noise, which made the calculation of electrochemical parameters difficult. The method of curve fitting was used to deal with electrochemical signals. Similar to the study by He and Zhang [36,37], we used Formulas (7) and (8) to fit the electric current–time curve and its integral electric quantity–time curve, respectively. Here two indexes were constructed to represent two different verification ideas. The parameter A, which represented the electric quantity index, was used to verify the influence of bacteria on the effective reaction area of the electrode from the perspective of the actual reaction rate of the electrode. The current index *i*_0_, which represented the constant current, was used to verify the influence of bacteria on the equivalent resistance in the electrochemical system from the perspective of the equivalent circuit.
(7)$$y=y_{0}+A\cdot e^{-\frac{x}{t}}$$
(8)$$y=A\cdot x^{\frac{1}{2}}+B$$

Consequently, as evident from Appendix A, interference by environmental noise was markedly reduced, and the fitting of the curves did not affect the original electrochemical results. Using this method, it became feasible to analyze changes in *i*_0_ and A within 200 s. As the current response in this study was the oxidation current, I < 0 and Q < 0. In fact, the current and electricity were not usually expressed as negative numbers, so the two indicators were taken as absolute values in the follow-up discussion. In addition, *i*_0_ showed an obvious decrease with an increase in bacterial concentration.

According to Formulas (7) and (8), *y*_0_ and *A* in the fitting result corresponded to *i*_0_ and *A*, respectively.

To verify the accuracy of our method, we performed the same experiment using multiple electrode systems (Appendix A). Figure 2A,B show the correlation among *A*, *i*_0_, and the bacterial concentration under the same electrode system. There was an obvious negative correlation among them. With an increase in bacterial concentration, the number of bacteria adsorbed on the electrode surface increased, resulting in a decrease in the actual electrode area S and an increase in equivalent resistance of the external circuit *R_s_*. According to Formulas (2) and (4), *A* and *i*_0_ would show a decrease. The electrochemical response index was linearly related to the bacterial concentration (*lg* value). Under the same electrode system, the relationship between the electrochemical response index and *E. coli* concentration was as follows:(9)$$\left|A\right|=-9.330\times 10^{-7}\times lgC+5.942\times 10^{-5}$$
(10)$$\left|i_{0}\right|=-5.907\times 10^{-7}\times lgC+2.908\times 10^{-5}$$
the *R*^2^ of Formulas (9) and (10) are 0.885 and 0.851, respectively.

Appendix A show the correlation between *A*, *i*_0_, and the bacterial concentration under different electrode systems. As evident, the different systems had a considerable influence on the electrochemical response. Due to the large difference in the actual reaction area (microscopic area) between the different electrodes, the effect of bacterial adsorption on the actual reaction area of the electrode was different, but the calculated error range remained below 30.0%.

This could be attributed to two reasons. First, the electrochemical response index had a better relationship with the adsorption position of *E. coli* on the electrode surface, which could be due to the “edge effect”, whereby the electric field strength at the edge of the electrode was higher than that at the center [38]. Thus, the adsorption of bacteria on the entire electrode surface was not uniform. Second, *E. coli* cells adsorbed on the electrode surface might agglomerate or overlap [30], leading to individual cells affecting the electrochemical response index to different degrees.

To the best of our knowledge, the electrochemical response indices A and *i*_0_ have not as yet been used for the quantitative detection of bacteria. Lebègue et al. [39] recently reported a quantitative method to detect bacterial concentration, which involved monitoring the collision frequency and adsorption of bacteria on the surface of an ultra-microelectrode. The detection accuracy of their method was, however, low, being two orders of magnitude different from the actual bacterial concentration. Moreover, the method was unable to differentiate between dead and live bacterial cells. Herein, the effect of dead bacteria and the error range of our method were assessed, and the results are shown in Figure 3 and Appendix A, respectively.

When different concentrations of bacteria were ultrasonically disrupted, and intracellular lysates were used as the test sample, *A* and *i*_0_ showed no obvious correlation with bacterial concentration, and A and i_0_ values were much larger than those for live bacteria (Figure 3A,B). This could be because the lysate may not be effectively adsorbed on the electrode surface, validating that our method was able to differentiate between live bacteria and intracellular lysates of dead bacteria. Similarly, when bacterial cells were fixed with glutaraldehyde, A and i_0_ showed no obvious correlation with bacterial concentration and were much higher than those for live bacteria (Figure 3C,D). To explain, after fixation, the two aldehyde groups of glutaraldehyde are bound to have cross-linked with the amino or imine groups on adjacent peptide chains, forming a dense network structure on the cell surface and considerably reducing the negative charge, consequently decreasing the migration rate in the electric field [40]. The experimental results showed that neither the cracked cell fragments nor the dead bacterial cells with complete cell structures could affect the equivalent resistance of the electrochemical system and the effective reaction area of the electrode. The reason behind this phenomenon was that the dead bacterial cells or cellular components could not migrate in the electrolyte and adhere stably on the electrode surface. To summarize, our method could distinguish dead bacteria (irrespective of them being intact or disrupted) from live bacteria. Only live bacteria can show a specific effect on the electrochemical response; intracellular lysates and cell structure did not affect the detection results.

The test results of the spiked samples (Appendix A) showed that although the difference in the actual area (microscopic area) of the electrodes results in greater dispersion of the test results of the different electrode systems, the relative standard deviations (RSDs) were all lower than 10%, which indicated that this method, though it used a different electrode system, still had certain stability and reproducibility. Due to the different micro surface areas of different electrodes, bacteria were not completely adsorbed on the electrode surface; the recovery rate was less than 100%. *E. coli* with a concentration of 10^8^ CFU/mL was selected, and the same electrode was used for intermittent detection and repeated verification, 10 times; the final RSD was 6.98%, which further indicated that the established sensor had certain stability and repeatability. There were few practical application scenarios where the bacterial concentration was higher than 10^9^ CFU/mL; these were not considered.

In summary, our method was able to rapidly detect *E. coli* within 5 min. As we used unmodified GCE in this study, the sensitivity of the method to lower bacterial concentrations was not sufficiently high. We compared the detection limit of our method with that of other chronoamperometric methods involving a functionally modified electrode [41,42,43,44,45]. Although the lowest detection limit was not outstanding, the detection method we reported had obvious advantages in terms of detection speed and difficulty of sensor preparation. The performance comparison of some *E. coli* electrochemical biosensors is listed in Appendix A. In addition, for severely polluted water environments and rapid quantification of *E. coli* solutions in the laboratory, this detection range was appropriate, and the detection of solutions with lower bacterial concentrations could be achieved by pre-concentration treatment.

### 3.3. Relationship between Bacterial Adsorption on the Electrode Surface and Bacterial Concentration

Ronspees and Thorgaard [30] reported that *E. coli* was adsorbed on the electrode surface with positive charge under the electric field, continuously hindering the oxidation of ions in the electrolyte on the electrode surface and consequently reducing the response current. Guanyue et al. [46] also showed that bacteria had a “blocking effect” on the response current and that this was related to the adsorption rate of bacteria on the electrode surface, adsorption position on the electrode surface, and longitudinal distance between the electrode and bacteria. However, the response current decreased to a certain extent when individual cells were adsorbed on the electrode surface. In addition, it is noteworthy that the blocking effect of each bacterial cell on the response current was not the same, resulting in differences in the reduction of the response current. In this study, we evaluated the relationship between the number of bacteria adsorbed on the electrode surface and changes in the total response current, establishing the relationship based on the principle of *E. coli* adsorption on the GCE surface.

We first performed scanning electron microscopy (Figure 4) to assess the adsorption state of *E. coli* on the GCE surface. Then, the number of bacteria adsorbed on the electrode surface was detected via fluorescence microscopy (Figure 5). Finally, differences in the bacterial concentration in the electrolyte before and after electrochemical detection were calculated to quantify the number of *E. coli* adsorbed on the electrode surface (Figure 6).

The GCE surface (Figure 4A) was smooth and neat, showing no bacterial cells or impurities. Short rod-like particles with a length of 2–3 μm were evident on the electrode surface after electrochemical detection (Figure 4B). Because the detection system was unmodified and all procedures were performed under sterile conditions, we strongly believe that these particles were in fact *E. coli* cells (even cell debris) that had been adsorbed on the GCE surface after electrochemical detection.

As shown in Figure 5, the higher the bacterial concentration, the higher was the number of *E. coli* cells adsorbed on the electrode surface. However, this bacterial adsorption was not uniform, which might be due to the roughness of the GCE surface at a microscopic level. This uneven adsorption could also be caused by the aforementioned “edge effect” [38].

In the present study, we also examined the electrochemical response of the Gram-positive strain *S. aureus* (Appendix A), which was found to be significantly different from that of *E. coli*. This could be attributed to the unique cell wall structure [47] and smaller cell size [48] of *S. aureus*. Ronspees and Thorgaard [30] reported that *Bacillus subtilis*, another Gram-positive bacteria, bounced off after collision on the surface of an ultra-microelectrode instead of showing continuous adsorption; this could be due to the electroosmotic flow in the electrolytic cell [49]. This partially explains why the collision state and adsorption efficiency of *S. aureus* on the electrode surface were quite different from those of *E. coli*. Therefore, the collision frequency, adsorption state, and adsorption efficiency of different kinds of bacteria on the GCE surface were also different, which leads to their different effects on electrochemical reactions; thus, it was theoretically feasible to use this method to specificity detect the bacteria.

Figure 6A, B show changes in the number and concentration of *E. coli* in the electrolyte before and after electrochemical detection. As evident, the higher the bacterial concentration, the higher the number of bacteria adsorbed on the electrode surface. Figure 6 further determined the quantitative relationship between the number of bacteria adsorbed on the electrode surface (i.e., the electrode area covered by bacteria) and the bacterial concentration based on Figure 4 and Figure 5. Further, different bacterial concentrations showed the same adsorption efficiency. The amount of negative charge on the surface of *E. coli* was basically the same, and the zeta potential was approximately −53.0 mV [30]. Therefore, the electrophoretic effect was the same under the same electric field.

In addition, by linear fitting the data in Figure 6A, we could calculate the following:(11)lg|ΔE|=0.976⋅lgC−1.589
the *R^2^* of Formula (11) is 0.997.

Further, according to Formulas (4), (9) and (11), we could calculate the following:(12)$$S=\frac{\pi^{\frac{1}{2}}}{2nFD_{0}^{\frac{1}{2}}C_{0}^{*}}\times \left(-9.559\times 10^{-7}\cdot lg\left|\Delta E\right|+5.790\times 10^{-5}\right)$$

Using Formula (12), we could determine that the number of *E. coli* cells adsorbed on the electrode surface *ΔE* was inversely proportional to the actual area S of the electrode, which conforms to the theoretical explanation and expected results.

In general, the greatest significance of Formula (12) was that through the change of parameters in the formula, the influence of bacterial concentration on the electrical signal and the reason for the influence of the process machine could be analyzed very intuitively. In addition, Formula (12) also indirectly verified the accuracy of Formulas (2) and (4). For example, Formula (2) described the relationship between the equivalent resistance of the electrochemical system and the bacterial concentration in the process. With the increase of the bacterial concentration in the detection solution, the equivalent circuit Rs in the detection system increased. On the premise that the transition voltage *E* remained unchanged, the current index *i*_0_ decreased with the increase of *Rs*. Another example was Formula (4), which described the relationship between the effective area of the electrode and the bacterial concentration. The increase of the bacterial concentration would increase the number of bacteria attached to the electrode surface, resulting in the decrease of the effective reaction area *S* of the electrode. Under the condition that other detection conditions remained unchanged, the constants or parameters such as *n*, *F*, *D_o_*_,_ *C_o_^*^*, and π remained unchanged, and the electrical index *A* decreased with the decrease of *S*. These formulae explained the experimental phenomena from the principle level, and intuitively reflected the influence mechanism of bacteria on electrical signals.

### 3.4. Influence of Humus on Test Results

Humus is widely distributed in nature and is formed by the decay of terrestrial animal and plant materials through the biological activities of microorganisms such as bacteria and algae in the soil environment [50]. These include the degradation products of carbohydrates, lignin, and proteins [51], which are generally divided into three categories: humin, HA, and FA. Dissolved humic acid (humic acid and fulvic acid) was rich in the water environment, accounting for 40–60% of DOM in the aquatic system [52]; fulvic acid (FA) was the main component of natural organic matter (NOM) [53]. In this experiment, FA dissolved in water was selected as the sample of common pollutants in water.

Hwang [54], Nagao [52] and Mir-zavand [55] selected humic acid with a concentration of 10 mg/L to simulate the river environment. In this experiment, 10 mg/L water-soluble FA was used as a distractor to observe the impact of pollutants in water on the detection results. The results showed that under the conditions of this experiment, FA had almost no effect on the detection results (Appendix A), which might be due to the fact that although humic acid had a negative charge when pH was greater than 5.5 [53], it could not generate a redox reaction on the electrode surface or affect the actual reaction area of the electrode by adsorption. Therefore, it could be inferred that the soluble humic acid in the water environment and small molecules that cannot undergo redox reaction cannot affect the detection results of *E. coli*.

## 4. Conclusions

In this study, we report a quantitative label-free method involving an unmodified GCE to detect live *E. coli*. Considering the characteristics of *E. coli* in a direct current electric field, bacterial cells were observed to migrate toward the GCE and were adsorbed on the electrode surface, which consequently hindered electron transfer and indirectly affected the electrochemical response. By monitoring this response over a period of time, the bacterial concentration could be determined. In addition to being simpler, faster, and more straightforward, our method was less affected by environmental noise and thus suitable for detecting 10^4^–10^8^ CFU/mL concentrations of bacteria. Compared with other similar studies, the electrode used in this study had not undergone any complex modification; it could be reused many times. In addition, the detection time was reduced to less than 5 min. This study analyzed the influence mechanism of bacteria on electrical signals. Compared with other studies focusing on the modification, preparation, and performance analysis of electrochemical biosensors, this study was more inclined to discuss and analyze the influence mechanism of bacteria on the electrochemical system.

As mentioned in the text, our method solved the shortcomings of the reported detection methods, did not need complex modification pretreatment, reduced the detection time to less than a few minutes, reduced the detection error, and could eliminate the interference of dead bacteria and humic acid. This study is expected to become an effective alternative to the plate counting method in the laboratory; thus, it is possible to realize the rapid estimation of the total number of bacteria in water after solving the challenge of adsorption efficiency of multi-strain complex systems in the future. We believe that our findings provide new ideas and a foundation for the development of robust detection technologies in the future.

## Figures and Tables

**Figure 1 biosensors-12-00845-f001:**
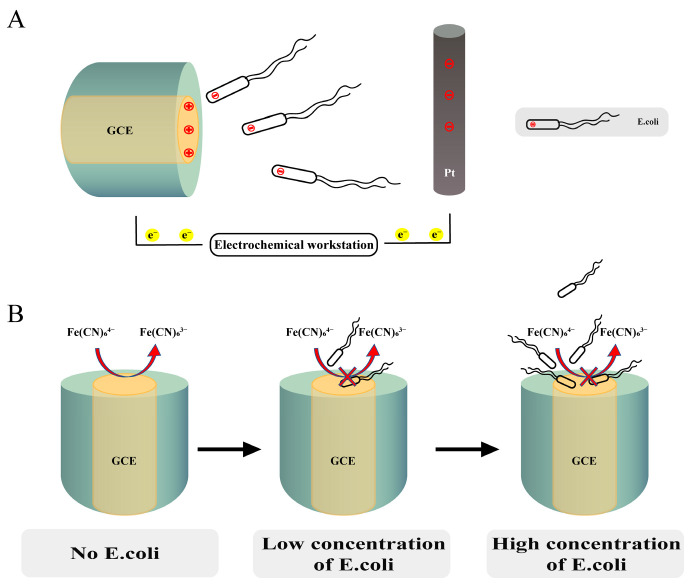
Schematic diagram of the proposed detection method. (**A**) *E. coli* migrates towards GCE in the electrolyte; (**B**) adsorption of *E. coli* on the glassy carbon electrode (GCE) surface, resulting in electron transfer being blocked.

**Figure 2 biosensors-12-00845-f002:**
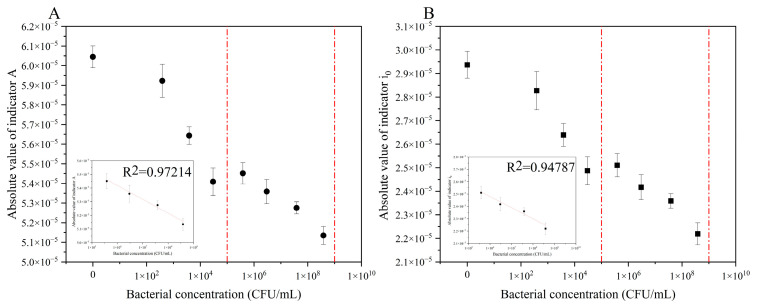
Electrochemical response of the same electrode system to different *Escherichia coli* concentrations. (**A**) electric quantity index; (**B**) electric current index. These two detection indicators were inversely proportional to the concentration of *E. coli*.

**Figure 3 biosensors-12-00845-f003:**
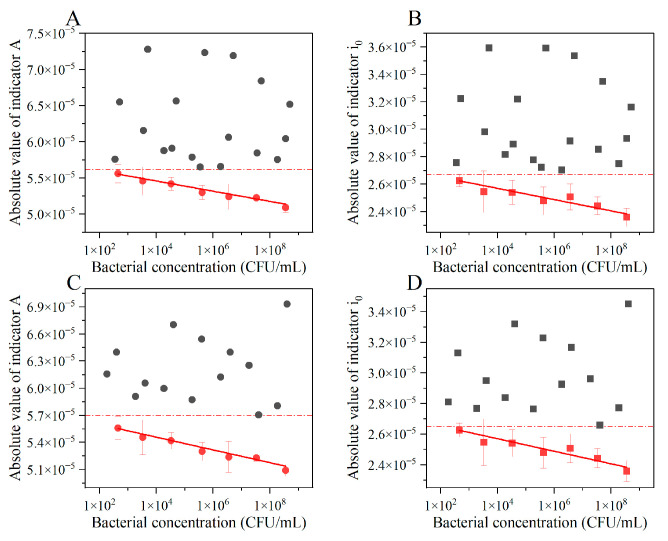
Influence of dead bacteria on the electrochemical response indices under different electrode systems. (**A**) Electric quantity index and (**B**) electric current index values when analyzing bacterial lysates after ultrasonic disruption; (**C**) electric quantity index and (**D**) electric current index values to detect bacterial structure after glutaraldehyde fixation. Black symbols indicate the concentration of original bacterial cells before the dead bacteria inactivation treatment, and red symbols indicate the concentration of viable bacteria; there was a significant difference between dead bacteria and live bacteria response indicators.

**Figure 4 biosensors-12-00845-f004:**
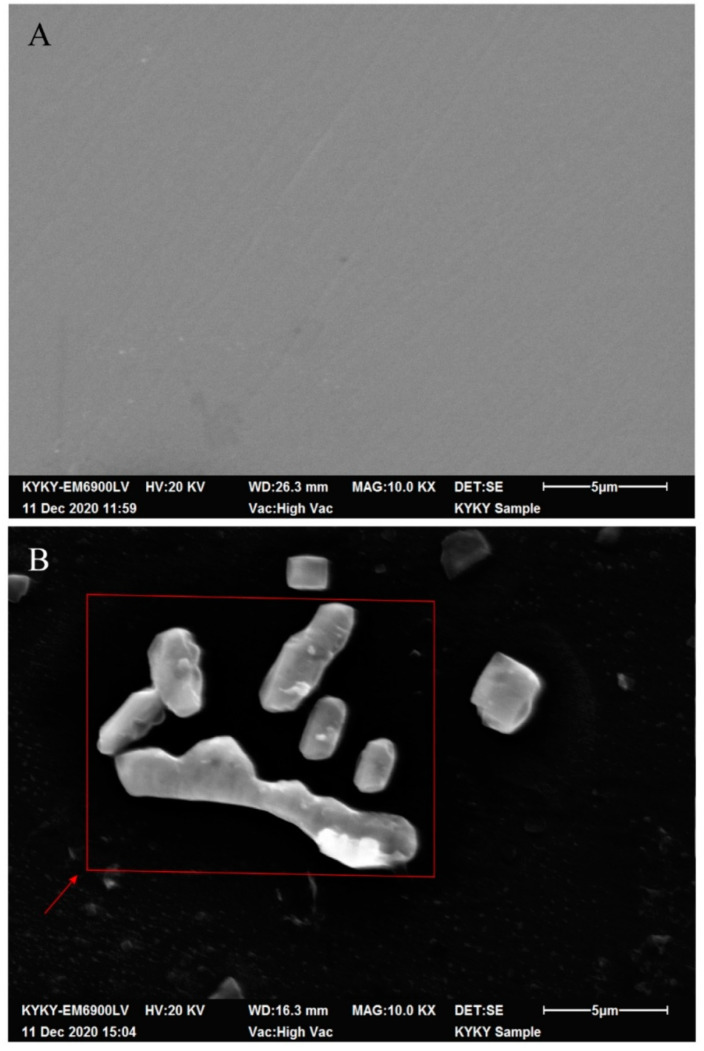
Scanning electron microscopy of the electrode surface after electrochemical detection. (**A**) The surface of the GCE after testing the sterile electrolyte was smooth and tidy; (**B**) after detecting the bacteria-containing electrolyte, the GCE surface had obviously adsorbed the bacterial cell structure.

**Figure 5 biosensors-12-00845-f005:**
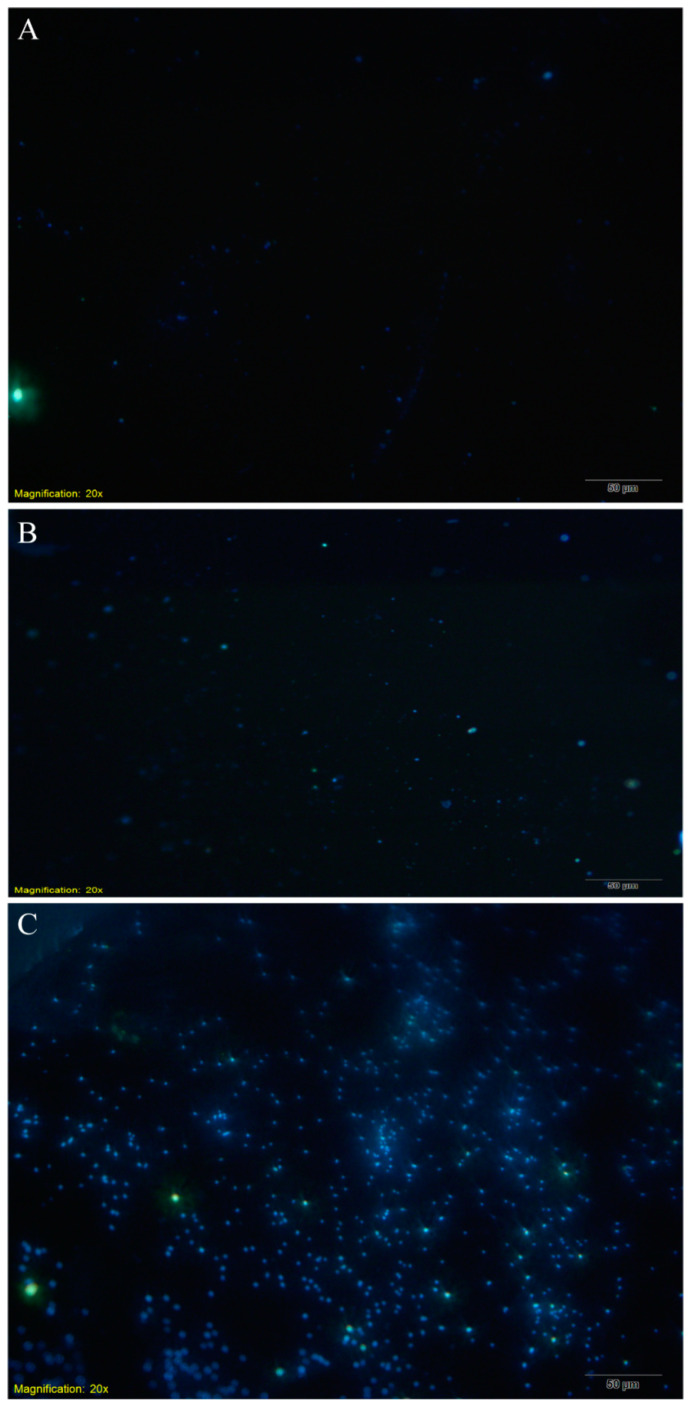
Fluorescence microscopy of the electrode surface after electrochemical detection. (**A**) The electrode surface after detection of sterile electrolyte; (**B**) the electrode surface after detecting electrolyte with bacterial concentration of 10^5^ CFU/mL; (**C**) the electrode surface after detecting electrolyte with bacterial concentration of 10^8^ CFU/mL. The yellow fluorescence represents impurity particles, and the blue fluorescence represents *E. coli* cells. The number of bacteria adsorbed on the GCE surface increases with the increase of the bacterial concentration in the electrolyte.

**Figure 6 biosensors-12-00845-f006:**
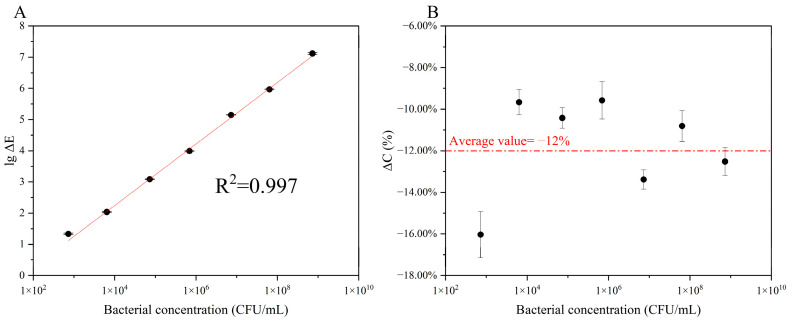
Changes in the bacterial concentration in the electrolyte before and after the test. The relationship between the changes in the number (**A**) and concentration (**B**) of the bacteria in the electrolyte before and after the detection and the initial concentration of *E. coli*.

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
