# Peer review of "Rapid Quantitative Detection of Live Escherichia coli Based on Chronoamperometry"

_biosensors, 2022, doi:10.3390/bios12100845_

Round 1
Reviewer 1 Report
The manuscript Rapid Quantitative Detection of Live Escherichia coli based on Chronoamperometry is interesting because offers a rapid quantitative detection of Escherichia coli (E. coli) for evaluating water and food safety, but poor to the analytical point of view. The auhtor need tor eprote the reproducibility, sensitivity and stability of their proposed system and The RSD o standard deviation in their graphs and figures.
Reviewer 2 Report
Authors should update their manuscript extensively with maintaining journal format. It can be considered for publication after major revision. Following are my suggestions and concern -
Comments:
1. The Abstract- was poorly and randomly written without maintaining the minimum standard/format. Authors should rewrite it. It should not contain unnecessary sentences that can be described in the Result discussion section.
Suggestion: What existing problem are you solving? What is your proposal? Method? Characterization? Result? Application? Future prospect?
2. “detection time still remains long” what does they want to say about this sentence?
3. Line 70 should merge with line 60.
4. Line 65-68 should merge within 40-50
5. INTRODUCTION should also be rewritten with proper format as well as clear conclusion at the last of each paragraph.
6. “In brief, we successfully achieved the label-free quantitative detection of E. 106 coli using two electrochemical indices, when the bacterial concentration was 104–108 107 CFU/mL, the calculated error range was <30%” Authors can describe these thing at the conclusion.
7. Why did the authors select initial potential as 0.6V. Did they optimize it?
8. How many GCE samples prepared to test one concentration or one sample prepared to test zero to high concentration?
9. Conclusion is too long. No standard format. Why do they need to use Refs in the CONCLUSION part too?
10. Why not to compare the performances in a table with other recently reported works?
11. What about the selectivity data? How did the authors confirm E.coli adsorption on the GCE substrate? How did they think, if similar analytes absorbed on the GCE substrate at the same potential?
What is the reason behind the negative error% in the table S-1
Round 2
Reviewer 1 Report
The authors answered in satisfactory way to the review request improving the mansucript
Reviewer 2 Report
Authors resolved my comments and concerns well. Now it can be accepted in the current form.